# An effluent pump family distributed across plant commensal bacteria conditions host- and organ-specific glucosinolate detoxification

Dor Russ [1,2], Connor R. Fitzpatrick[1,2], Chinmay Saha [1,2], Theresa F. Law[1,2], Corbin D. Jones[1,3,4], Daniel J. Kliebenstein [5] & Jeffery L. Dangl [1,2] ✉

In nature, plants recruit a diverse microbial community, the plant microbiome, that is distinct from the surrounding soil community. To understand the forces that shape the plant microbiome we need to characterize the microbial traits that contribute to plant colonization. We used barcoded mutant libraries to identify bacterial genes that contribute to the colonization of a monocot and a eudicot host. We show that plant colonization is influenced by dozens of genes. While some of these colonization genes were shared between the two host plant species, most were highly specific, benefiting the colonization of a single host and organ. We characterized an efflux pump that specifically contributes to Arabidopsis shoot colonization. This efflux pump is prevalent across Pseudomonadota genomes yet benefits the bacterial association with only a small subset of *Arabidopsis thaliana* accessions. Leveraging genomic diversity within *Arabidopsis thaliana*, we confirmed that specific glucosinolate breakdown products are detoxified by this family of efflux pumps. The broad prevalence of this efflux pump family suggests that its members contribute to protection of commensal bacteria from collateral damage of plant glucosinolate-based defense responses to herbivores and necrotrophic pathogens.

Plants host diverse microbial communities that play critical roles in plant development, nutrition, and resistance to disease[1–4]. Plants can contribute to the selection of those communities from the surrounding environment through a myriad of chemical and physical means: production of antimicrobials; deposition of root exudates into the soil; competition for nutrients; physical barriers, and more[5–10]. While the microbial communities associated with different plants share similarities at high taxonomic levels, the community composition diverges at lower taxonomic ranks[11–14]. Moreover, plants recruit distinct communities to different organs[15]. Our knowledge of processes and proteins dictating plant or species-specific microbiome assembly remains limited. To fill this gap, we must define the microbial genes determining colonization and persistence in the plant microbiome[16]. Understanding the forces that shape plant microbiomes will enhance our ability to predict the behavior of microbial communities and to harness those communities for sustainable agriculture.

Different approaches can be taken to study microbial traits that help microbes colonize plants. Studies comparing the genomes of

[1]Department of Biology, University of North Carolina at Chapel Hill, Chapel Hill, NC, USA. [2]Howard Hughes Medical Institute, University of North Carolina at Chapel Hill, Chapel Hill, NC, USA. [3]Integrative Program for Biological and Genome Sciences, University of North Carolina at Chapel Hill, Chapel Hill, NC, USA. [4]Department of Genetics, University of North Carolina at Chapel Hill, Chapel Hill, NC, USA. [5]Department of Plant Sciences, University of California Davis, Davis, CA, USA. ✉e-mail: dangl@email.unc.edu

plant-root-associated bacteria versus those found in other habitats found enrichment of genes related to carbohydrate utilization and higher predicted growth rate potential[17,18]. Furthermore, in planta meta-transcriptome profiling of root-inhabiting bacteria revealed an upregulation of genes involved in energy production relative to free-living bacteria[19]. Assimilation of specific glucosinolates can drive differences in bacterial communities between closely related hosts[20]. However, direct interrogation of microbial genes required for persistence in plant habitats remain rare. In recent years, whole genome, high-resolution bacterial mutant libraries have enabled genetic screens to infer causality in bacterial–host microbiome interactions. These were deployed to understand antibiotic resistance, phage resistance, metabolite assimilation, gut colonization and more[21–23]. Due to the complexity of mutant library application on plants, colonization screens to date have been typically performed in simplified, artificial systems, often limited to a single host and/or organ[24–27]. These experiments identified genes important for plant colonization, including genes that help bacteria evade plant immunity, control bacterial motility, and regulate different metabolic processes[24–26]. Yet, the diversity of bacterial genes important for long-term colonization is unknown as are the diversity of mechanisms by which they encode colonization traits are for individual plant hosts and organs. To address this knowledge gap, we deployed a high-resolution screen to reveal genes defining host and organ specificity in the plant microbiome.

We identified ~250 plant-association genes through a genome-wide approach using bacterial mutant libraries of two different robust plant-colonizing bacterial strains growing on either the monocot *Brachypodium distachyon* or the eudicot *Arabidopsis thaliana*. We demonstrate that the majority of colonization genes affect the association with specific organs or specific hosts with only a few genes benefiting the colonization of different organs in both plant species. We focused on genes inferred to function in bacterial adaptation to plant defense and immunity and identified two types of organ-specific efflux pumps, one that protects the bacteria against an Arabidopsis-shoot-derived glucosinolate and another that likely protects the bacteria against an as yet unidentified Arabidopsis-root-derived compound. Searching for functional orthologs of the efflux systems in the genomes of a large bacterial isolate collection, we found that the shoot-specific efflux pump (*ef90*) is prevalent among Arabidopsis colonizing commensals. The *ef90* pump protects the bacteria from a toxic compound expressed in the leaves of small subset of *A. thaliana* accessions. We leveraged recombinant inbred lines (RIL) and genome-wide association (GWA) analysis to identify the glucosinolate breakdown product sulforaphane as the antibacterial compound detoxified by *ef90* encoded pumps in *A. thaliana* Col-0. The *ef90* pump was previously defined as *saxF* in hemi-biotrophic leaf pathogenic strains of *Pseudomonas syringae*, where it was proposed to protect the pathogen from glucosinolates produced during necrotrophic growth[28]. Our data prompt re-evaluation of this hypothesis.

## Results

### *A. thaliana* and *B. distachyon* assemble distinct microbial communities

Plant hosts actively recruit their associated microbiota. General colonization requirements would select for traits that enable generalized colonization across species and organs. Conversely, distinct species and strain-level microbiome assembly associated with different plant organs and hosts would suggest specific interactions selecting for specialized bacterial traits. We used two highly diverged model plants, the eudicot *A. thaliana* Col-0 and the monocot *B. distachyon* BD21 grown in a single wild soil that we have sampled extensively over the last 15 years (Mason Farm, Chapel Hill, NC) ("Methods"). After 6 weeks of growth in this soil, we harvested roots and shoots and characterized community assembly using 16S rRNA sequencing ("Methods"). We

found that the two organs of the two plants assembled bacterial communities distinct from the soil community and from each other (Fig. 1a). We noted that the betaproteobacteria genus *Paraburkholderia* exhibited robust colonization of both organs of both host species and was readily detectable in soil (Supplementary Fig. 1). This genus had high and stable relative abundance in all five habitats. Genetic tools are available for *Paraburkholderia bryophila* MF376[29] (hereafter MF376), a genome-sequenced isolate previously cultured from *A. thaliana* Col-0 roots from plants growing in the same wild soil[17]. The evolutionary distance between the two hosts, their distinct microbiomes, and the ability to define the molecular colonization requirements of MF376 on both hosts represented an ideal opportunity to compare and contrast the bacterial genes that affect association with divergent plant species and different organs.

### Dozens of bacterial genes affect association with plants

We deployed a TnBarSeq screen[29] to identify MF376 genes that regulate host association across different organs of Arabidopsis or Brachypodium. Each bacterial clone in the library has one gene disrupted by a randomly barcoded *mariner* transposon. We inoculated the MF376 TnBarSeq mutant library onto *A. thaliana* Col-0 and *B. distachyon* BD21 seeds after sowing in a soil-like gnotobiotic system[30] to mimic the natural colonization process (Fig. 1b; "Methods"). We harvested soil and plants after 6 weeks of growth, separated the aboveground (shoots) from below-ground (roots) organs, and sequenced the population of barcodes ("Methods"). We calculated the relative abundance of mutations in each gene[21]. We identified barcodes in 5962 genes (90% of the genome) and a total of 80,047 unique barcodes with an average of 50,544 barcodes per sample and ~8.5 unique barcodes (insertions) per gene ("Methods"; Supplementary Fig. 2a).

Mutations could, in principle, affect bacterial colonization of the different habitats in different ways (Fig. 1b): no effect (indifferent genes, gray); reduction of barcode abundance in all habitats (general plant association genes, blue), or in only some plant tissues compared to soil (organ-specific, green, or host-specific, turquoise); or reduction of the barcode abundance in soil compared to plant tissues (negative plant-association genes, red). We defined from these data plant-association genes as those genes where clones with mutations in them were depleted from plant tissue compared to the soil (Fig. 1c, blue dots and left inset) and negative plant association genes as those where clones with mutations in them were enriched in plant tissue (Fig. 1c, red dots and right inset). Characterizing the genes that affected association with different plant tissues (Fig. 1c, Supplementary Fig. 2b, and Supplementary Data 1) provided mechanistic insight into the principles and mechanisms of MF376 colonization.

### Most MF376 colonization genes are host- and organ-specific

Surprisingly, we found that most colonization genes benefited the association of MF376 with a specific host and/or organ. Thus, while we identified 244 genes as *context-specific* colonization genes, only seven were identified as *general* colonization genes that benefited the association with all four tested plant tissues. We showed that 210 of the 251 plant-association genes were host- and organ-specific (first four columns of Fig. 2a). Only a few genes were host-specific and required for colonization of both organs (Fig. 2a, columns 5 and 10). Likewise, a few genes were required for colonization of the same specific organ in both species (Fig. 2a, columns 6 and 9). Few to no genes benefited different organs in the divergent species, i.e. benefit in Arabidopsis root and Brachypodium shoot, but not elsewhere (Fig. 2a, columns 7 and 8). More genes specifically benefited the association with plant shoots (Fig. 2a, columns 1, 3, and 6; 179 genes) than roots (Fig. 2a, columns 2, 4, and 9; 48 genes). This result implies that the selection on bacterial genes in plant shoots is more stringent than that in root (versus selection in the soil). This inference is consistent with the observation that shoot communities exhibit lower similarity with soil than root

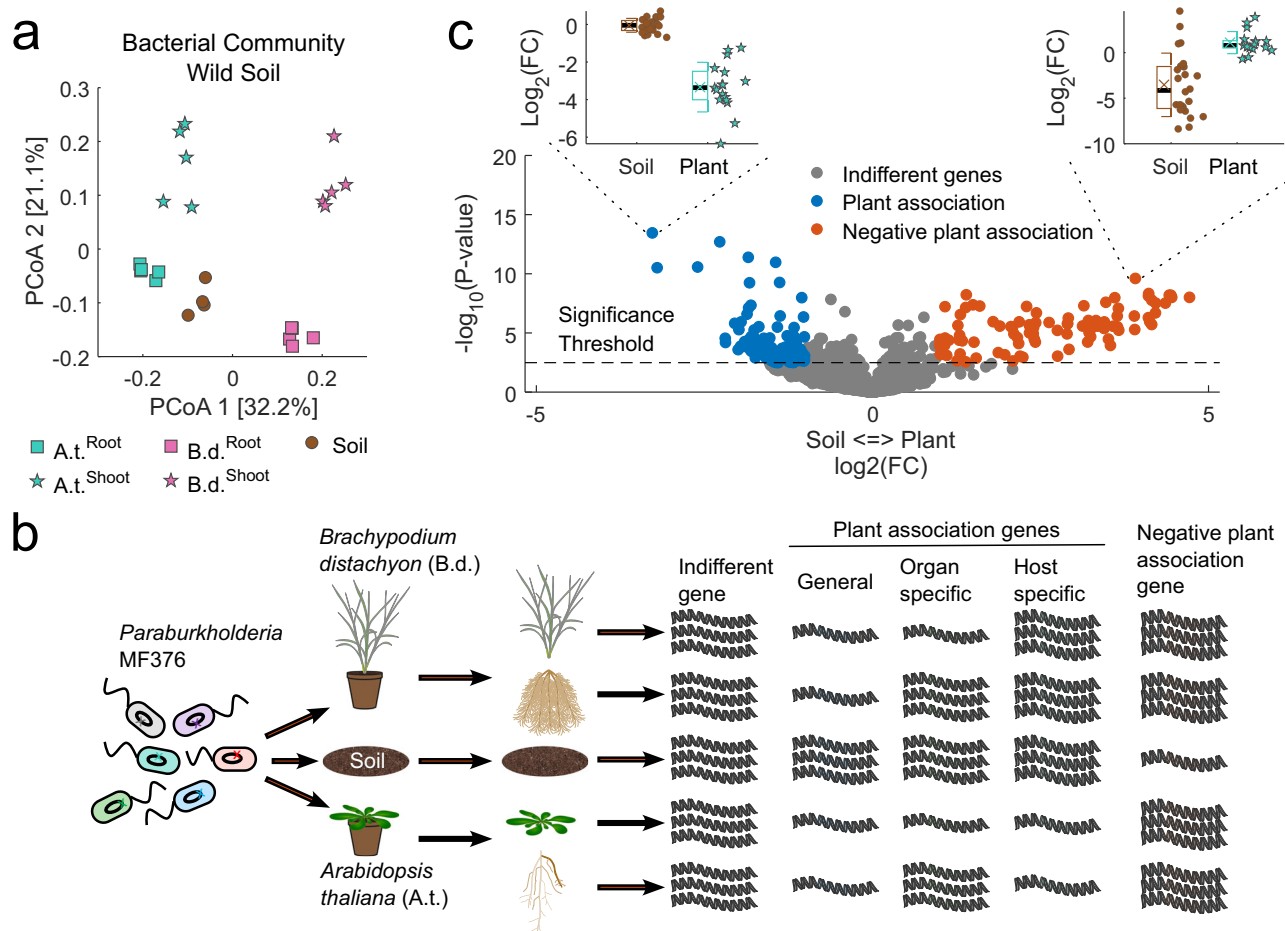

**Fig. 1 | Identification of bacterial genes required for full colonization of two divergent plant host species. a** Principal coordinate analysis of the bacterial community inhabiting wild soil (Mason Farm soil, brown circles, $n = 4$) and the roots (circles, $n = 5$) and shoots (stars, $n = 5$) of two plant hosts: *Arabidopsis thaliana* (A.t. turquoise) and *Brachypodium distachyon* (B.d. magenta). The ordination is based on Bray–Curtis distance and shows that the microbiomes of the two organs of the two hosts are different from one another and from the soil. **b** A schematic representation of the experimental approach and questions addressed. A TnBarSeq mutant library of the robust plant colonizer *Paraburkholderia bryophila* MF376[29] (bacteria, left side; each color represents a different mutant) was inoculated onto autoclaved potting soil at the time of sowing of *Arabidopsis thaliana* (A.t.) and *Brachypodium distachyon* (B.d.) seeds, and left to colonize the plants for 6 weeks. The roots, shoots, and soil were sampled. The abundance of each mutant was identified by amplicon sequencing. Individual mutations in MF376 genes could have no effect on a particular mutant's abundance in the different environments

(indifferent gene, gray), could reduce the mutant's abundance in all or some plant hosts and organs (mutants in plant association genes, blue, green, and turquoise), or could reduce the mutant's abundance in soil compared to its abundance in plants (negative plant association gene, red). **c** Volcano plot comparing mutant abundances in the soil ($n = 24$ biologically independent samples) to their abundances in association with plant tissue ($n = 20$ biologically independent samples). Fold changes (FC) and $P$ values for individual mutants were calculated based on soil samples (insets, brown circle) and plant samples (insets, turquoise star). Differentially abundant mutants were statistically different (significance threshold, Linear model, FDR corrected $P$ value <0.05) and were either depleted in association with plant tissue (plant association, blue, $\text{Log}_2(\text{FC}) < -1$, representative gene in left inset) or enriched in association with plant tissue (negative plant association, red, $\text{Log}_2(\text{FC}) > 1$, representative gene in right inset). All other mutants were defined as indifferent (gray). Boxplots show the median (black line), mean (colored x), first and third quartiles (box), and one standard deviation (whiskers) of treatment.

communities (Fig. 1a). Negative plant-association genes were usually shared among multiple organ–host combinations, 48 of the 104 negative plant association genes were shared among three organ-host combinations (Fig. 2a, red and column 11; the Brachypodium root was the outlier because only a few bacterial genes significantly affected association with it, either positively or negatively). The high number of small-effect association genes (Figs. 1c and 2a), together with the robust colonization by Paraburkholderia of both host plants (Supplementary Fig. 1) and the high species and/or organ context-dependent specificity of colonization genes (Fig. 2a) emphasizes the notion that plant association is a complex, multigenic, phenotype driven by evolution to maximize context-dependent adaptive flexibility.

We analyzed the functional categories of these genes (Fig. 2b and Supplementary Data 1) using Clusters of Orthologous Genes (COG). GO processes such as carbohydrate and amino-acid metabolism, energy production, transcription, and cell wall integrity (Fig. 2b) are enriched

across plant association genes for all plant organs. These results recapitulated, in general, findings from previous studies[19,25,26]. We found that genes related to cell motility and to intracellular trafficking and secretion were enriched among negative association genes (see full list of enriched categories in Supplementary Data 2). The genes driving the signal in the intracellular trafficking and secretion category were, in fact, structural genes in the bacterial flagellum apparatus essential for cell motility. These results suggest that the loss of MF376 motility enhances colonization. While this result differs from a previous study[25], we note that the two studies surveyed colonization requirements of different focal strains in very different experimental systems.

**Two efflux pump systems promote association with either *A. thaliana* Col-0 roots or shoots**

We identified two efflux systems that confer organ and host specificity and likely interface with plant defenses. First, we identified three genes

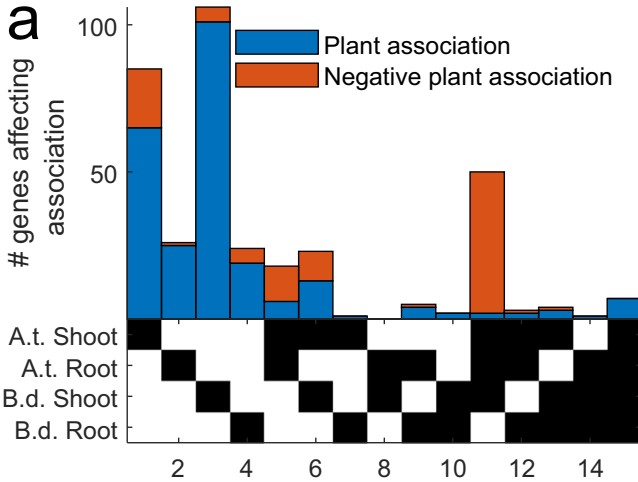

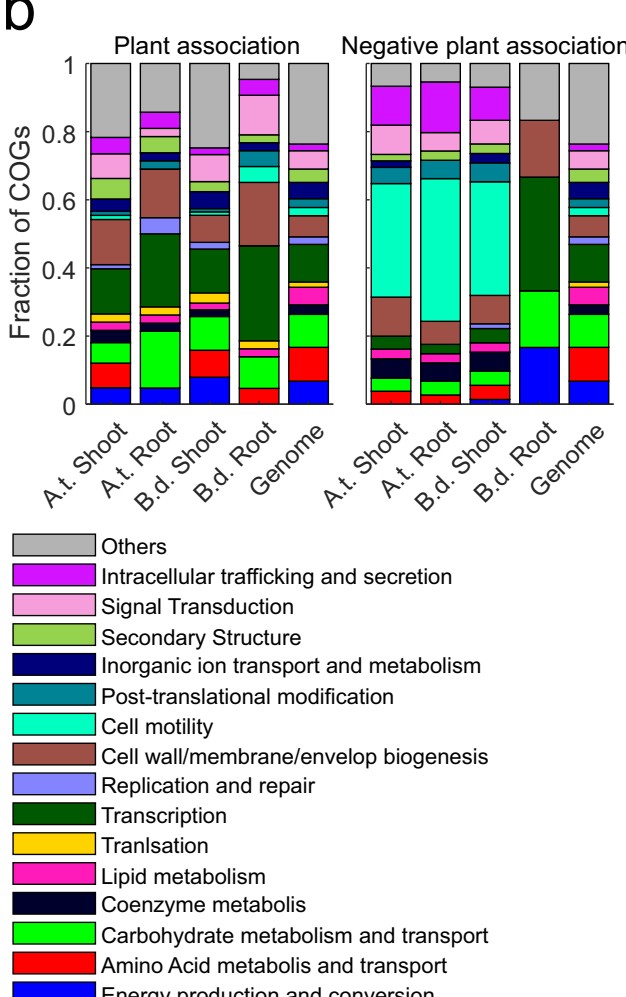

**Fig. 2 | Most MF376 colonization genes affect specific host and organ association. a** An upset plot of genes that affect plant association. Genes that positively (blue) or negatively (red) affect association with different host and organ combinations (bottom, black squares). Column numbers are shown at bottom and are referred to in the text. See full list of genes in Supplementary Data 1. **b** Functional analysis of bacterial genes that affect plant association. Abundance of gene functional categories within plant association genes (left) and negative plant association genes (right) compared to their abundance in the bacterial genome (right-most bar in both panels) according to COG classification.

in a predicted operon at MF376 locus IDs H281DRAFT_03189, 03190, and 03191 (with 5, 16, and 13 insertions per gene respectively) that encode proteins with high homology to the three subunits of an efflux pump from the resistance, nodulation, and cell division (RND) super-family: an inner membrane protein (Locus ID 03190), a periplasmic protein (03189), and an outer membrane protein (03191) (hereafter, the *ef90* operon). Insertions into the three genes comprising this operon had similar effects on association. RND efflux pumps are membrane-bound complexes that protect bacteria from anti-microbials by specifically binding deleterious compounds and trans-porting them out of the bacterial cell[31]. RND systems are prevalent among Gram-negative bacteria and each such system has a specific repertoire of ligands it can bind and transport[32]. All three efflux pump genes (Fig. 3a, green, top panel) specifically benefited the association of MF376 with *A. thaliana* Col-0 shoots. Second, we identified a similar efflux system at MF376 loci IDs H281DRAFT_05196, 05197, and 05198 (with 13, 26 and 19 insertions per gene, respectively) that specifically benefited the association with *A. thaliana* Col-0 roots (Fig. 3a, purple, second panel). Neither of these two putative efflux systems benefited the association of MF376 with either *B. distachyon* BD21 shoots or roots (Fig. 3a).

RND efflux pumps often remove antimicrobials to confer anti-microbial resistance. Logically, our findings suggested that *A. thaliana* Col-0 produces at least two different antimicrobials, one in the shoot and one in the root (Fig. 3b), that affect MF376 fitness. We hypothesized that MF376 encodes the two organ-specific efflux pumps to remove putative growth-inhibiting compounds to retain full fitness during plant colonization (Fig. 3b, left). To expand our understanding of commensal plant-association genes in general, and the role of RND efflux pumps in colonization in particular, we repeated the mutant library screen in Arabidopsis and Brachypodium across both organs with a second TnBarSeq library of *Pseudomonas simiae* WCS417 (hereafter, WCS417), a model gammaproteobacteria commensal[33] ("Methods"; Supplementary Fig. 3a and Supplementary Data 3). The WCS417 results qualitatively recapitulated our findings using MF376 (Supplementary Fig. 3b). We again identified two RND organ-specific efflux systems that promoted the association with either *A. thaliana* Col-0 shoots (Loci IDs PS417_13190, and 13195 with 53 and 7 insertions per gene respectively) or roots (Loci IDs PS417_01045, and 01050 with 7 and 53 insertions per gene respectively) with (Fig. 3b, right; Supplementary Fig. 3c).

We tested the requirement of the *ef90* operon for fitness on *A. thaliana* Col-0 shoots using a competition assay (Fig. 3c; "Methods"). We first engineered parental MF376 derivatives by knocking resistance to either chloramphenicol (Chl^R wild-type (WT)) or tetracycline (Tet^R WT) into an intergenic region (see "Methods"). We next deleted the entire *ef90* operon from each of the parental MF376 derivatives to create scarless derivatives called Chl^R d90 and Tet^R d90, respectively (see "Methods"). We then competed parental WT strains against reci-procally antibiotic-resistant mutant d90 strains in macerated *A. thali-ana* Col-0 leaf extract[34,35] by mixing equal cell numbers of Chl^R WT with Tet^R d90 or Chl^R d90 with Tet^R WT. We chose leaf extracts because we hypothesized that pre-formed antimicrobials would be at higher con-centration than in apoplastic wash fluids. We quantified the growth of each strain after plating the overnight mixed culture on Luria Broth (LB) agar plates supplemented with either tetracycline or chlor-amphenicol (see "Methods"). We found that while the WT strains were unaffected by the presence of Col-0 macerated leaf extract, the *ef90* deletion, MF376 d90, resulted in complete growth inhibition when in competition with WT MF376 (Fig. 3c). The parental WT strains and the d90 mutants grew to similar levels when competed in non-extract-supplemented medium or in root extract (Supplementary Fig. 4). We also tested an independent WCS417 mutant allele with a disrupted inner-membrane subunit (*ef90* ortholog) isolated from a gridded, sequence-indexed copy of the WCS417 TnBarSeq library[25] (Locus ID PS417_13190) (hereafter, WCS417-d90). We measured the bacterial

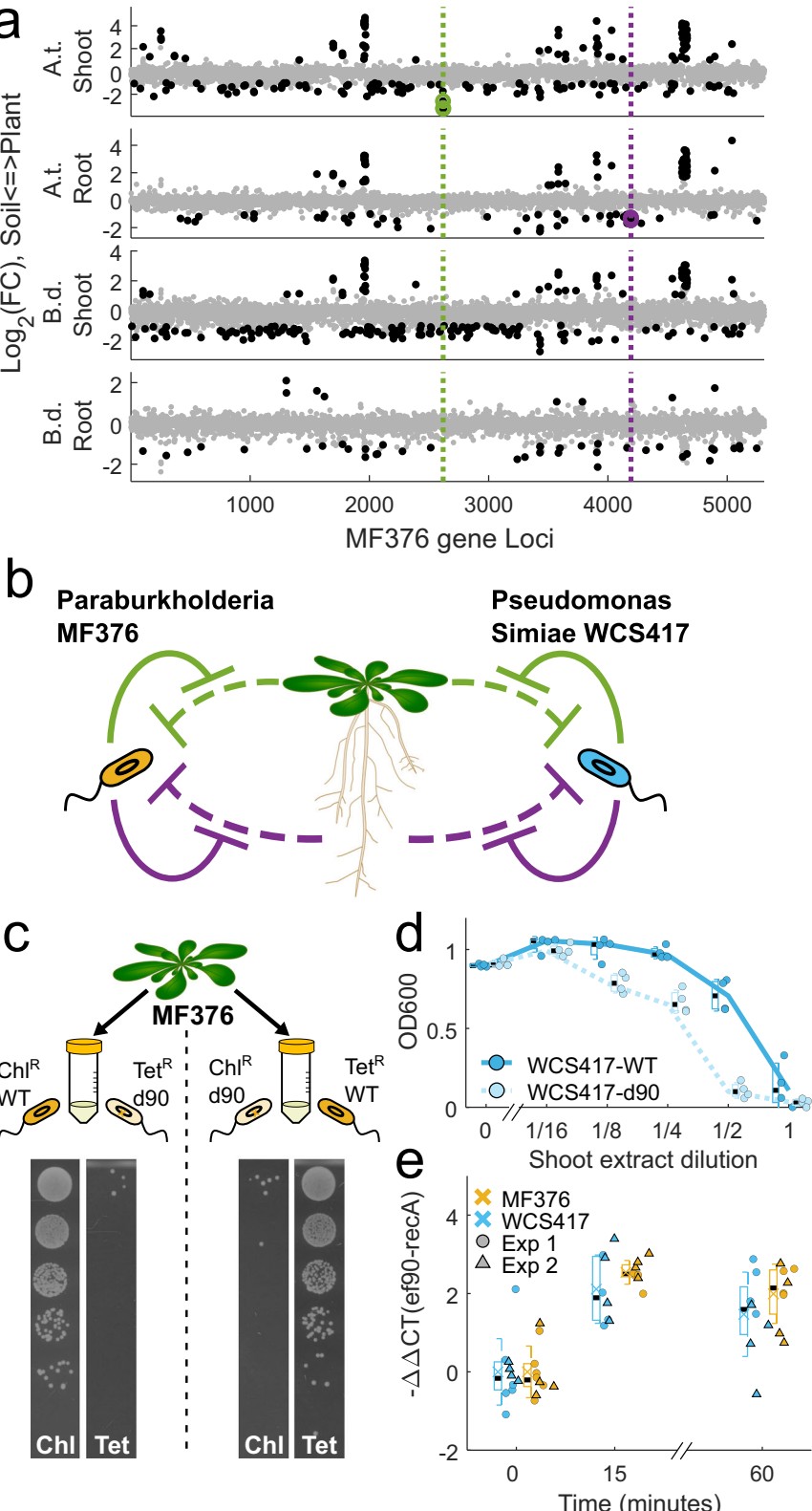

growth of the WT WCS417-WT strain and the mutant WCS417-d90 separately in macerated *A. thaliana* Col-0 leaf extract diluted in medium (Methods). We found that macerated leaf extract inhibited the mutant WCS417-d90 growth when the extract was diluted 1:8 into medium. By contrast, the growth inhibition of the WCS417-WT strain was only evident at a 1:2 dilution (Fig. 3d). Overall, these results demonstrate that the genes that encode the *ef90* efflux pumps in both

MF376 and in WCS417 confer fitness against compounds found in macerated *A. thaliana* Col-0 shoots. The analogous functions of *ef90* efflux pumps in MF376 and WCS417 suggests a common mechanism determining their requirement for association with a particular organ, leaves, of a specific host, Arabidopsis, is defined.

Bacterial RND efflux pump gene expression is often induced when the bacteria are exposed to a cognate compound[36,37]. We leveraged

**Fig. 3 | Organ-specific efflux pumps are required for full colonization of *Arabidopsis thaliana* Col-0 and fitness in leaves. a** Locus position in the MF376 genome of statistically significant plant association mutations (black dots, association genes below the gray dots, negative association genes above). Organ-specific efflux pump systems benefit the association with Arabidopsis shoots (green) or roots (purple) but not with other organs and host. Log2(FC) of mutant abundance in association with plant tissue compared to soil is plotted against their location on the genome. **b** Our data suggest a model in which Arabidopsis produces two different antimicrobial agents, one from the shoot (green) and one from the root (purple), that can inhibit bacterial growth. To reach full colonization MF376 and WCS417 express different organ-specific efflux pumps that remove the plant-derived compound and protect the bacteria. **c** Reduced fitness of mutant *Paraburkholderia bryophila* MF376 d90 in an extract of macerated Arabidopsis leaves measured in a competition assay. A Chloramphenicol-resistant (Chl^R) *Paraburkholderia bryophila* MF376 clone with scarless deletion of the shoot association efflux pump (d90, light orange) was competed against a Tetracycline resistant (Tet^R) wild-type clone (WT, dark orange) overnight in macerated Arabidopsis leaves (top). Bacterial growth was quantified by plating serial five-fold dilutions of 1:1000

diluted overnight culture (left to right) on agar plates containing differentiating antibiotics, demonstrated that, while the wild-type clone grows well in macerated leaves, the mutant bacteria did not grow. A reciprocal experiment (bottom) confirmed these results. **d** Reduced fitness of *Pseudomonas simiae* WC417 d90 in macerated Arabidopsis leaf extract. Final $OD_{600}$ of overnight cultures of either wild-type *Pseudomonas simiae* WCS417 (WCS417-WT, dark cyan, solid line, circles, $n = 3$ biologically independent cultures) or a mutant in the homologous shoot association efflux pump (WCS417-*d90*, light cyan, dashed line, triangles, $n = 3$ biologically independent cultures) in medium supplemented with diluted macerated Arabidopsis leaf extract (shoot dilution). **e** Expression of the bacterial shoot association efflux pump, *ef90*, was induced following infiltration into the Arabidopsis apoplast (at $10^7$ cfu/ml Expression of *ef90* in either *Pseudomonas simiae* WCS417 (cyan) or *Paraburkholderia bryophila* MF376 (orange) 15 and 60 min after infiltration into Arabidopsis leaves (For each $n = 8$ biologically independent leaves in 2 independent experiments). In both (**d**, **e**), boxplots show median (black line), mean (colored x), first and third quartiles (box) and one standard deviation (whiskers) of each treatment.

that regulatory mode to test bacterial transcriptional response to *A. thaliana* Col-0 shoots in planta. We infiltrated MF376 and WCS417 into leaf apoplasts of 6-week-old plants and quantified the expression levels of each strain's *ef90* inner membrane subunit using reverse transcriptase quantitative polymerase chain reaction (RT-qPCR; Fig. 3e; "Methods"). We found that within 15 min of infiltration *ef90* expression was induced fourfold in both MF376 and WCS417. That induction implies that upon leaf infiltration the bacteria are exposed to the *A. thaliana* Col-0-derived compound that inhibits growth of the *d90* mutant.

## Homologs of the *A. thaliana* Col-0-associated, shoot-specific *ef90* efflux pump are prevalent among Arabidopsis-associated bacteria

Given the specificity of the MF376 *ef90* fitness benefit for growth on macerated leaf extracts of *A. thaliana* Col-0, and the presence of a similar specific efflux pump in WCS417, we asked if these efflux pumps were a common feature of bacteria strongly associated with Arabidopsis. We used a 185-member collection of genome-sequenced bacteria isolated from Arabidopsis roots[4,13,17]. The RND gene family of efflux pumps is widespread among Gram-negative bacteria. We identified the genes that encode all RND efflux pump inner-membrane subunits (also called RND transporter, and hereafter, RND pump) in Gram-negative bacteria in our isolate collection (Fig. 4a; "Methods") because this subunit determines the compound specificity of RND efflux pumps[38]. RND efflux pumps are highly abundant in all Gram-negative bacteria in our collection, regardless of taxonomic class, with an average number of 11 genes per isolate. We generated a neighbor-joining tree of all the genes based on sequence similarity (Fig. 4b; "Methods"). The tree is diverse; genes from bacteria of all RND efflux pump classes are distributed along it. Based on the most recent common ancestor of the functionally tested RND efflux pumps from MF376 and WCS417 (orange and cyan dots), we identified clades of genes potentially associated with *A. thaliana* Col-0 shoots (green, hereafter shoot clade) or roots (purple, hereafter root clade). While the root clade is deep and diverse, the shoot clade is shallow and monophyletic, implying that genes in that *ef90* ortholog group perform similar functions and are diverging.

We found that most of the Pseudomonadota in our collection encode an *ef90* ortholog in the shoot clade (Fig. 4c; 78% of Alphaproteobacteria, 100% of Betaproteobacteria, and 91% of Gammaproteobacteria; Supplementary Data 4). Most isolates encode only one gene in that clade (Fig. 4d). Five of the seven bacteria that encode two genes in the shoot clade belong to Variovorax, where a second copy of the gene has diverged from the rest of the tree after gene duplication (Fig. 4e; long diverging

orange Comamonadaceae branch). Because the shoot clade genes are clustered with orthologs from bacteria of the same family, we tested whether gene expression increased in *A. thaliana* Col-0 leaf extracts from a single representative strain of each major family (Fig. 4e; "Methods"; Supplementary Data 5). We found that *ef90* ortholog expression is induced following exposure to macerated leaf extract from *A. thaliana* Col-0 in five of seven tested, clade representative strains (Fig. 4f).

The prevalence of *ef90* orthologs in our collection also led us to a wider search for *ef90* orthologs in bacterial genomes[17]. We found that *ef90* is also widespread among bacteria isolated from other plants (Supplementary Fig. 5 and Supplementary Data 6). Importantly, we noticed that many plant pathogens, including *P. syringae* pv. tomato DC3000, carry *ef90* orthologs (Supplementary Fig. 6). That ortholog in DC3000, known as *saxF*, is a clade A RND efflux pump that protects the bacteria against the antibacterial aliphatic glucosinolate breakdown product sulforaphane[28,39]. The existence of this third example of an *ef90* pump playing a role in colonization, the shallow monophyletic shoot clade, the high prevalence of a single copy of the shoot pump gene among Arabidopsis-associated bacteria and the transcriptional induction by *A. thaliana* Col-0 macerated leaf extract suggest that many *saxF/ef90* orthologs play roles in removing shoot-derived defense toxins, likely aliphatic glucosinolate breakdown products[28], produced in *A. thaliana* Col-0 shoots.

## The plant-derived antimicrobial detoxified by *saxF/ef90* is rare among angiosperms and even across *A. thaliana* accessions

Our screen identified mutants that affected the association of MF376 and WCS417 with *A. thaliana* Col-0 shoots but had no effect on colonization of *B. distachyon* BD21. Given the large phylogenetic distance between these two species, it is not surprising that the bacteria associated with these diverged taxa would have efflux pumps tuned to different host antimicrobials like glucosinolates. Given the specificity of glucosinolate-based defenses mainly to the Brassicales[40], we asked how conserved the observed leaf-derived inhibitory function was across finer plant evolutionary scales. Using the competition assay described in Fig. 3c, we surveyed additional eudicot host plants from different families, the solanaceae *Solanum lycopersicum* (*S.l.*), the legume *Medicago truncatula* (*M.t.*). These extracts did not inhibit MF376 d90, consistent with expectations, because the aliphatic glucosinolates detoxified by SaxF are limited to the Brassicaceae and possibly the Capparaceae (Fig. 5a). Shoot extract from *Capsella bursa-pastoris* (*C.b.*), a close relative of Arabidopsis from the Brassicaceae that makes a completely different set of aliphatic glucosinolates[40], also did not inhibit MF376 d90 growth, nor induce expression of *ef90* (Fig. 5a and Supplementary Figs. 7 and 8). To further refine specificity

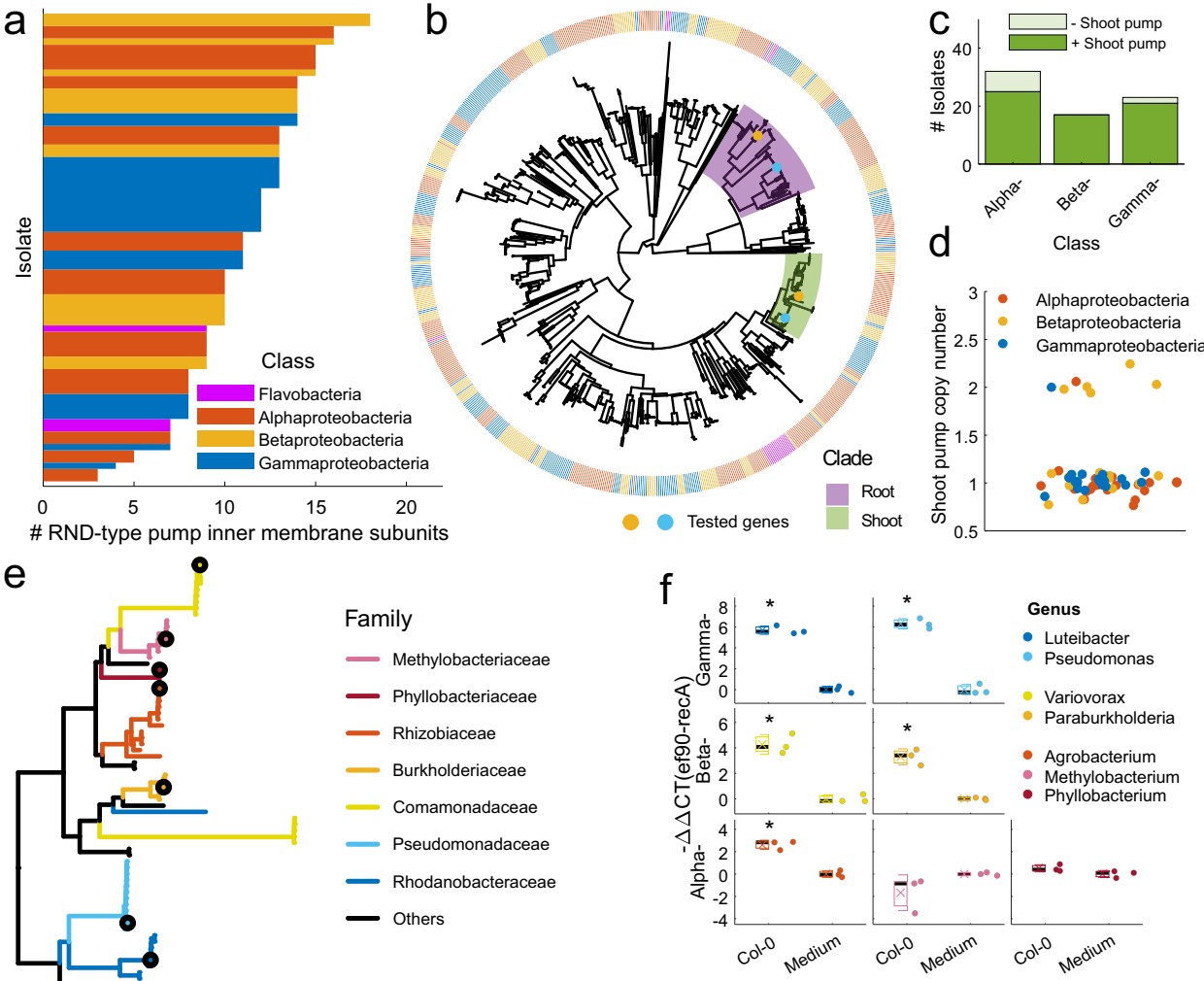

**Fig. 4 | Homologs of *ef90* are prevalent among *Arabidopsis thaliana* colonizing Pseudomonadota. a** The number of RND-type efflux pump system inner-membrane subunits in each Gram-negative bacterial genome of a 185 member genome sequenced isolate collection of bacteria collected largely from *Arabidopsis thaliana* roots. **b** A gene tree of all inner-membrane subunits in the 185-member isolate collection (bacterial class in the outer circle) based on sequence similarity (see "Methods"), and position of the functionally tested genes from *Paraburkholderia bryophila* MF376 (orange dot) and *Pseudomonas simiae* WCS417 (cyan dot). The clades related to the shoot (green) and root (purple) RND efflux pump systems were identified by the most recent common ancestor (MRCA) branch of the functionally tested genes. The functionally tested shoot-specific genes are in a shallow clade while the root-specific genes are in a deep and divergent clade. **c** In

the genomes of each of the tested classes of Pseudomonadota, the vast majority of isolates encode an *ef90* homolog from the shoot clade (78% of Alphaproteobacteria, 100% of Betaproteobacteria and 91% of Gammaproteobacteria). **d** Most isolates encode only a single copy of the shoot clade *ef90* homolog, but seven of the 63 strains encode two copies. **e** A subtree of the shoot-specific clade colored by bacterial family marking one representative from each major family (family with more than two isolates, black circle). Five of the seven duplicated genes in Fig. 5d are from a divergent group of genes in five *Variovorax* strains (Comamonadaceae, orange long branch). **f** Expression of *ef90* is induced in *Arabidopsis thaliana* Col-0 macerated leaf extract compared to bacteria in medium for five of seven representative strains (**e**) from all classes (*n* = 3 biologically independent cultures, Two sided *T* test *P* value <0.05).

of the production of the compound detoxified by saxF/ef90, we assessed seven accessions that represent a broad swath of *A. thaliana* diversity[13]. The MF376 d90 mutant was only inhibited by *A. thaliana* Col-0 extract (Fig. 5b). These results suggested that the beneficial effect of the MF376 efflux pump is narrow and possibly limited to only a fraction of Arabidopsis accessions.

## The saxF/ef90 efflux pump protects commensal bacteria from shoot-derived aliphatic isothiocyanates

We leveraged population genomics tools to dissect the genetics of the production of the antimicrobial detoxified by saxF/ef90. First, we exploited an established panel of 98 RILs of a cross between Col-0 and the non-inhibiting accession Ler-1[41,42]. We divided the phenotypes observed in the RIL population into three groups (Fig. 5b): Non-toxic (Ler-1-like behavior, magenta) did not inhibit the WT or the d90

mutant; d90-toxic (Col-0-like behavior, turquoise) inhibited only the d90 mutant; and a novel transgressive phenotype (yellow) that inhibited both the WT and the d90 mutant (WT-toxic; Supplementary Fig. 9a and Supplementary Data 7). Three broad loci exhibited strong linkage to the d90- and WT-toxic phenotypes (Supplementary Data 7). The homology between saxF and d90 (Supplementary Fig. 6) suggested that the Col-0-derived compound that inhibits the d90 mutant is an aliphatic glucosinolate breakdown product[28]. Consistent with this observation, each of the three peaks in the RIL analysis include a locus that exhibits presence/absence variation between Ler-1 and Col-0 affecting the production of an enzyme in glucosinolate metabolism (Fig. 6a, gray lines; see below). The RIL loci encompass wide genomic regions. We therefore refined our genetic mapping to test if these glucosinolate loci may be the cause of the d90-toxic and WT-toxic phenotypes using 92 select accessions from the *A. thaliana* 1001

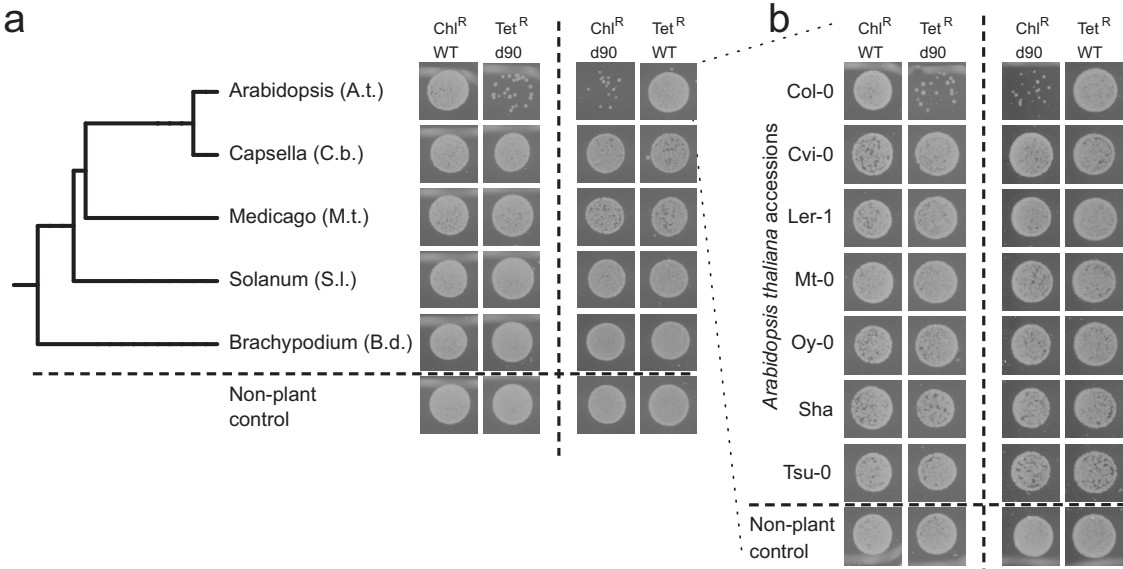

**Fig. 5 | The shoot-derived antibacterial compound is host-genotype specific.**
**a** Competition assays, as described in Fig. 3, between wild-type *Paraburkholderia bryophila* MF376 (WT) and a deletion mutant in the shoot association *ef90* efflux pump (d90) growing in a 1:1 mix of medium and macerated leaf extract of diverse plant host species (**a**; phylogenetic tree) or different *Arabidopsis thaliana* accessions (**b**). Each plant host was tested in two competition assays: chloramphenicol-resistant wild type, Chl[R] WT, against Tetracycline-resistant mutant, Tet[R] d90 (left), and the reciprocal competition Chl[R] d90 against Tet[R] WT (right). Competition results were measured by plating 1:1000 diluted overnight cultures on agar plates containing differentiating antibiotics. Only *Arabidopsis thaliana* Col-0 accession inhibited the mutant's growth.

genomes project[43]. Six out of the 92 accessions expressed the d90-toxic response; these accessions were all isolated in Germany and the Netherlands. Another five expressed the WT-toxic response from diverse sources across Eurasia, which suggests that the novel transgressive WT-toxicity observed in the RILs is found naturally in some wild accessions (Supplementary Fig. 9b and Supplementary Data 8). We performed a GWA analysis comparing the d90-toxic and non-toxic accessions (Supplementary Data 9) and observed a remarkable convergence with the RIL results. The single-nucleotide polymorphism (SNP) with strongest association colocalized with the first peak in the RILs analysis (Fig. 6a, b, ESP peak) and resides within the promoter of the gene epithiospecifier protein (ESP, AT1G54040). Whether or not this is the causal SNP remains to be determined, but we did correlate its presence to decreased expression of ESP (Supplementary Fig. 9c).

The ESP protein plays a key role in glucosinolate breakdown. Upon foliar damage following herbivory or necrotrophic pathogen infection in Brassicaceae, glucosinolates are broken-down by myrosinases to one of two possible bio-active products in an ESP-dependent manner. First, ESP can promote the formation of a nitrile moiety instead of the isothiocyanate moiety that is spontaneously formed when EPS levels are low or absent[44] (Fig. 6c). If the glucosinolate has a terminal alkene, ESP will create an epithionitrile. Second, natural genetic variation at the other two RIL peaks encode enzymes that epistatically interact with ESP to differentiate Col-0 and Ler-1 glucosinolate products: methylthioalkylmalate synthase (MAM also known as the GSL-Elong locus), where Col-0 contains a copy of *MAM1* (AT5G23010) that produces longer chained glucosinolates compared to *MAM2* in Ler-1; and 2-oxoglutarate-dependent dioxygenases (AOPs), where Ler-1 shoots, but not Col-0 shoots, express an active AOP3 (AT4G03050) that converts a methylsulfinyl (MSO) glucosinolate into a hydroxyl glucosinolate[45] (Fig. 6c).

The parental accessions, Col-0 and Ler-1, produce 4MSO-ITC and 3OHP-Nitrile, respectively, as their dominant chemotypes[46] (Fig. 6c, thick lines). The hybrid nature of the Col-0 x Ler-1 RILs population includes lines that by recombination shuffle the *ESP*, *MAM* and *AOP* loci to encode non-parental combinations of glucosinolate metabolic enzymes and diverse breakdown products. This leads to eight possible glucosinolate-derived active compounds (Fig. 6c). Of the breakdown products, 3OHP-ITC and 4OHB-ITC go through a rapid spontaneous cyclization to form 1,3-Oxazinane-2-thione and 1,3-Oxazepane-2-thione, respectively[47]. Bacterial inhibition by leaf extracts from specific lines in the Col-0 × Ler-1 RIL population (Fig. 6a and Supplementary Fig. 9a) suggests that three of those products (4MSO-ITC, 3MSO-ITC, and 1,3-Oxazepane-2-thione) are toxic to MF376 and are detoxified by the *saxF/ef90* efflux system. We tested commercially available isothiocyanates to validate that the *saxF/ef90* efflux system protects MF376 against 4MSO-ITC (Fig. 6d; Sulforaphane) and 3MSO-ITC (Fig. 6e; Iberin) as predicted. A fourth product (1,3-Oxazinane-2-thione) is specifically found in the RIL genotypes and accessions that have WT toxic extracts and is predicted to be highly toxic. We speculate that this compound may bypass the saxF/ef90 bacterial resistance and may be responsible for the WT-toxic phenotype, but it is not commercially available. Note that all of the lines that express a functional ESP, and thus produce nitriles, yield non-toxic extracts.

## Discussion

Plant microbiomes affect plants in natural ecological settings and in agriculture[2,48]. Yet, despite a growing knowledge base about the distinct microbiomes that inhabit different plants and plant organs, our understanding of the forces that drive microbiome assembly and function remains limited. Here, we show, first, that species- and organ-specific genes dominate the functional landscape of colonization and, second, that diversity in a class of antimicrobial efflux pumps contributes significantly to colonization by commensals that must contend with a diversity of host-produced glucosinolate metabolites deployed upon tissue damage.

We functionally surveyed the genomes of two plant colonizing bacteria, *P. bryophila* MF376 and *P. simiae* WCS417, for genes that affect the association with the above- (shoot) or below- (root) ground tissues of the two distantly related host-plant species *A. thaliana* Col-0 and *B. distachyon* BD21, we found ~250 genes that promote association with these plants. We demonstrated that a super-majority of plant-

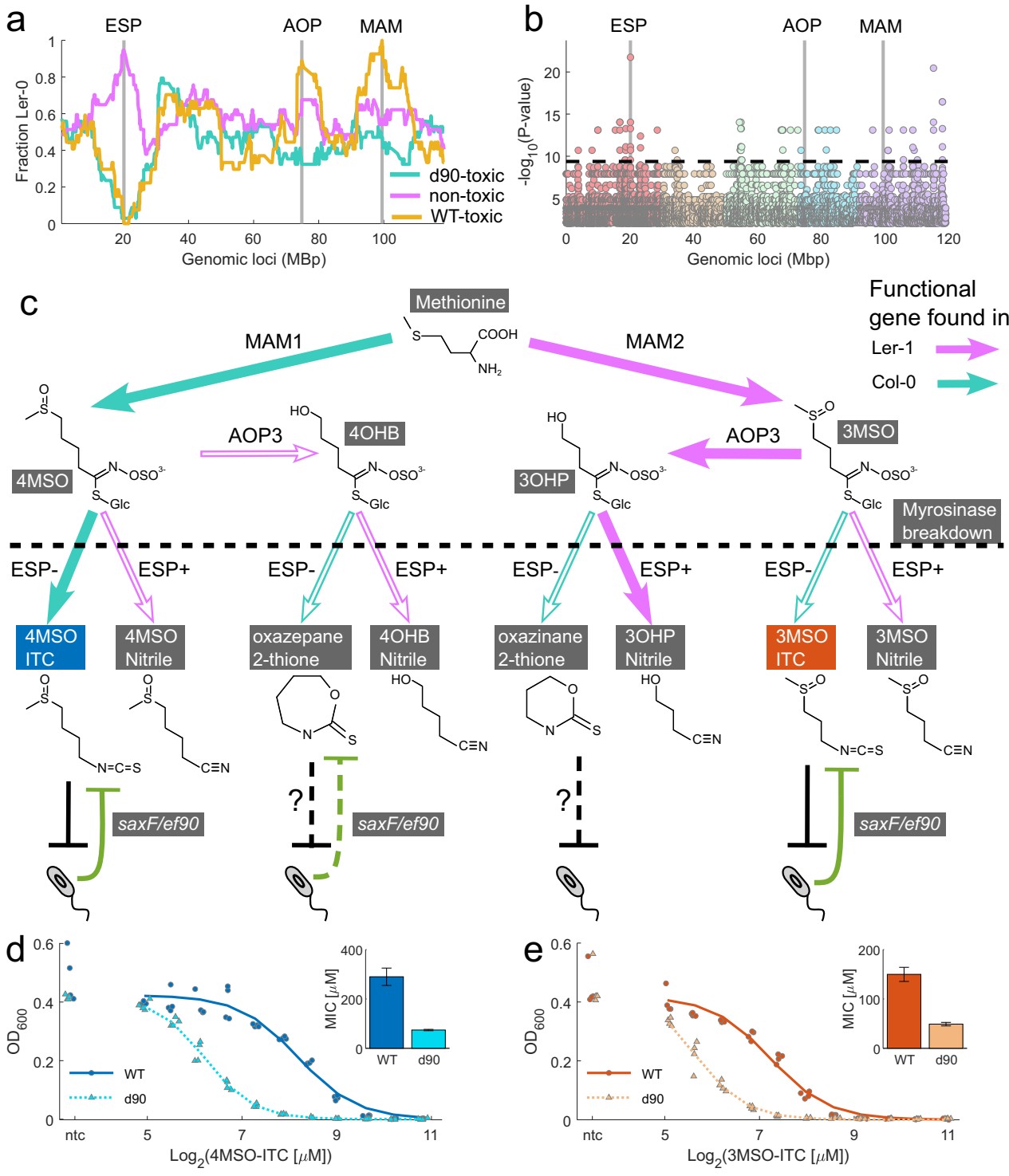

broadly. It is noteworthy that the roots of both Col-0 and Ler-1 produce nearly all nitrile glucosinolates, and no glucosinolate isothiocyanates[49] (see below), possibly explaining the shoot specificity of the *ef90* operon mutants.

association genes in these strains are organ- and host-specific. We focused on two bacterial efflux systems that exemplify such specificity, benefiting the association with either the roots or the shoots of *A. thaliana* Col-0, but not *B. distachyon*. Detailed analysis of one of these efflux systems, *saxF/90*, demonstrated that the plant-association benefit of the shoot-specific *saxF/ef90* operon is host genotype-specific, even within *A. thaliana*, and that orthologs of *saxF/ef90* are prevalent among Arabidopsis colonizing Pseudomonadota. Further genomic and functional analyses found that the *saxF/ef90* pump detoxifies specific aliphatic glucosinolate isothiocyanate break-down products (and not other phytoalexins, Supplementary Fig. 10) whose presence is polymorphic across *A. thaliana* and the Brassicaceae more

The aliphatic isothiocyanates 3MSO-ITC (Iberin) and 4MSO-ITC (Sulforaphane) that are detoxified by saxF/ef90 are present only in the shoots of a limited number of geographically restricted *A. thaliana* accessions, including Col-0. This observation is consistent with the diversity of glucosinolate chemotypes defined across this species[46]. Glucosinolate diversity has been associated with specific activity against diverse herbivores and necrotrophic pathogens[50]. Specific glucosinolates alter interactions with a diversity of biotic organisms

**Fig. 6 | The saxF/ef90 efflux system protects a commensal bacterial strain against glucosinolate breakdown products from *Arabidopsis thaliana* Col-0 leaves. a** Genetic mapping of antibacterial production in *Arabidopsis thaliana* Col-0 leaves using recombinant inbred lines (RILs) from a Col-0 × Ler-1 cross. The fraction of Ler-1 alleles at any given locus is plotted along the five chromosomes concatenated. There were three phenotypic RIL groups (Supplementary Fig 9a and Supplementary Data 7): Non-toxic (Ler-1-like, magenta); d90-toxic (Col-0-like, turquoise); and a novel transgressive WT-toxic phenotype (yellow). The d90 toxic phenotype maps to Col-0 alleles in an interval that includes the *ESP* locus (first vertical gray line). The WT-toxic phenotype maps to Col-0 alleles at the *ESP* region in combination with two additional loci carrying Ler-1 alleles in regions around the *AOP* and *MAM* loci (vertical gray lines). **b** Manhattan plot of genome-wide association (GWA) mapping for the ability to specifically inhibit the d90 mutant across 87 natural *Arabidopsis thaliana* accessions (Supplementary Fig. 9b and Supplementary Data 7 and 8). The analysis contrasts six accessions that inhibited d90 growth (Col-0-like phenotype) with 81 non-inhibiting accessions (Ler-1-like phenotype). The strongest linkage above a Bonferroni-corrected EMMAX *P* value <0.05

threshold (horizontal dashed line) is to a SNP mapped in the promoter of the *ESP* gene. The scale is identical to (**a**). **c** Combinatorial glucosinolate metabolic pathway in the Col-0 × Ler-1 RIL population. This metabolic map presents only steps that differentiate Col-0 (turquoise) from Ler-1 (magenta). Thick arrows represent the parental Col-0 and Ler-1 pathways; empty arrows represent novel allelic combinations present across the RIL population. The major glucosinolate defense catabolite produced in each genotypic class are shown below with isothiocyanates predicted or shown to be detoxified by the *saxF/ef90* efflux system denoted by the green inhibitory arrows. **d**, **e** Dose response curves showing the growth of wild type and the d90 mutant along a gradient of either 4MSO-ITC (**d**, Sulforaphane; blue, $n = 4$ biologically independent cultures), or 3MSO-ITC (**e**, Iberin; red, $n = 4$ biologically independent cultures). $OD_{600}$ measurements of the wild type (circles) and mutant (triangles) bacteria are fit to a decreasing Hill function (solid and dashed lines respectively) to find the compound concentrations that inhibit 50% of the growth (IC50). Inset: the minimal inhibitory concentration (MIC) defined as the IC50 of the compounds. Error bars represent the 95% confidence intervals of the estimated coefficients from the model fitted to the data.

and their structural diversity is likely being shaped by these combinatorial interactions; there net effects in any given leaf microhabitat including interactions affecting herbivores, pathogens and now commensals[51–53]. Conversely, the plant-associated commensal bacterial genomes we surveyed carry an average of 11 RND transporter genes per genome. The widespread distribution of closely related *saxF/ef90*-homologous operons in the genomes of plant-associated bacteria indicates that commensal bacteria encounter 3MSO-ITC (Iberin) and 4MSO-ITC (Suforaphane), or a structurally related product often enough for the pump function to be widely distributed and retained across evolutionary time. This may arise from the widespread albeit scattered presence of these compounds across the Brassicaceae[40]. Alternatively, the cost of maintaining the *saxF/ef90* operons might be so low that diverse plant-associated bacteria retain them due to the likelihood that they may encounter Brassicaceae hosts that do produce these aliphatic glucosinolates.

While we observed enrichment of *saxF/ef90* across the genomes of Arabidopsis-associated bacterial isolates compared to non-plant associated isolates, we did not observe enrichment across the genomes of Arabidopsis-colonizing bacteria compared to genomes of bacteria isolated from other plant species (Supplementary Fig. 5). Closely related plants can produce a variety of secondary metabolites that can function as antimicrobials with variation in tissue-specificity of expression and variation in toxin production within a species[54,55]. We speculate that *saxF/ef90* pumps are functionally important for removing these toxins from commensal bacteria across a wide range of plants. Plant-associated bacteria would be expected to benefit from carrying an array of efflux pumps to allow them to adapt to the unpredictable diversity of antimicrobial compounds they are likely to encounter when colonizing diverse plants.

Glucosinolates and their breakdown products differentially influence herbivory depending on whether the herbivores are specialists or generalists. But our understanding of the role of glucosinolates in shaping the plant microbiome is still limited. Some commensals can utilize specific glucosinolates as nutrient sources. Thus, utilization of allyl-glucosinolate derived isothiocyanates by bacteria[20,40], and plant ER body-associated myrosinases linked to non-host microbe resistance[56] can affect plant microbiome assembly. While the production of 3MSO-ITC (Iberin) and 4MSO-ITC (Sulforaphane), substrates of the *saxF/ef90* pump (Fig. 6d), is not widespread among the tested *A. thaliana* accessions (Fig. 5b and Supplementary Fig. 9b), they are produced by numerous widespread Brassicaceae species that can be tested in the future, including significant crops like Broccoli[57,58]. In addition, RND efflux pumps can have a repertoire of ligands[32,38], and the selection for retention and diversification of the *saxF/ef90* efflux pumps may be augmented by additional selective agents, potentially unrelated plant defense metabolites.

Interestingly, 3MSO-ITC (Iberin) and 4MSO-ITC (Sulforaphane) are products of glucosinolate breakdown by myrosinase following tissue damage[59]. The sampling of our in planta bacterial mutant library screen included plant tissue damage, known to lead to near complete conversion of the intact glucosinolates to the isothiocyanates within approximately 1 min[60]. It remains unclear how commensal bacterial confront such glucosinolate breakdown products in natural settings. Some glucosinolates can be found on the leaf surface, though their function there is not clear and there is evidence for steady state endogenous isothiocyanates expressed in the Col-0 accession[61,62]. Moreover, foliar microbial communities are likely under constant threat of collateral damage due to occasional bursts of a variety of glucosinolate breakdown products resulting from the many sources of mechanical damage plants face in nature. Indeed, herbivory and necrotrophic pathogen attack alter the plant microbiome by unknown mechanisms[63], potentially linked to glucosinolate breakdown[20]. Importantly, the taxon with highest prevalence of *saxF/ef90* pumps (Supplementary Fig. 5), Pseudomonas, is also the most enriched following herbivore attack of the glucosinolate producing *Cardamine cardifolia*[63]. The high prevalence of *saxF/ef90* among plant commensal bacteria and the enrichment of Pseudomonas upon herbivory suggest that *saxF/ef90* protects bacterial microbiota strains from collateral damage of the plant response to such challenges.

More broadly, the efflux systems we defined that contribute to host and organ specificity (Fig. 3) represent essential new details in our understanding of how a commensal bacterium determines host and organ specificity. Antimicrobial resistance in general, and efflux systems specifically, are established mechanisms for niche adaptation[64]. Antimicrobial resistance conferred, for example, by efflux systems, protects human pathogens from antibiotics applied in clinics, and can protect plant pathogens from plant-derived toxins[28,35,65,66]. These examples reinforce the concept that microbiome assembly, while largely deterministic, is also largely context dependent. Moreover, the high abundance of efflux pumps among Arabidopsis colonizers implies that other accessions and other plant species produce different repertoires of antimicrobials that select for other pumps. In addition, efflux pumps can also have a role in homeostasis management, siderophore efflux and more. Overall, our work demonstrates a role for efflux pumps in the host- and organ-specificity of plant commensal bacteria colonization.

In our in planta experiments testing the relative fitness of the MF376 *d90* mutants, we observed no growth inhibition compared to WT bacteria, even in wounded Arabidopsis leaves where glucosinolates are typically activated. Several factors may explain this outcome. First, glucosinolate activation is a dynamic process where exposure to herbivores or necrotrophic pathogens leads to continuous activation and breakdown. As such, our experimental wounding of *A. thaliana*

leaves may have resulted in a transient response insufficient to reach inhibitory concentrations during our experimental timeframe. Second, and perhaps more significantly, bacteria often possess redundant defense mechanisms against plant antimicrobials. This aligns with the findings of Fan et al.[28] that individual mutations, including saxF, exhibit prominent loss of detoxification function phenotypes in vitro but have negligible in planta effects. In this example significant susceptibility only manifests in a quadruple mutant (saxA, saxD, saxF, saxG). Therefore, while our d90 mutant demonstrates increased sensitivity to purified glucosinolates like sulforaphane or shoot extract in vitro, complementary efflux systems or detoxification pathways might provide sufficient protection against physiological concentrations of structurally diverse glucosinolates encountered in the plant environment. Further research is required to dissect this complex response.

Finally, the high specificity of plant-association genes and the plethora of genes and mechanisms that affect plant-association should inform methods to tailor therapeutic bacteria developed for the field[48]. Our results show that the presence of a particular bacteria trait, the saxF/ef90 efflux pump, can have profound effects on colonization success across even within a species. This high specificity potentially poses a challenge for application of microbial interventions across different cultivars and across different crop species, but this high specificity might also may allow opportunities for fine scale tailoring of microbial strains or synthetic communities to specific plant host genotypes.

## Methods

### Plant material
The following plant species were used in this study: *A. thaliana* Col-0, *B. distachyon* BD21, *S. lycopersicum* Microtom, *M. truncatula* R108-1, *C. bursa-pastoris*. In addition, we tested 98 RILs coming from a Col-0 × Ler-1 cross (Supplementary Data 7) and 92 other *A. thaliana* accessions (Supplementary Data 8).

### Microbiome analysis in wild soil
**Soil sample and plant growth.** Soil was collected from Mason Farm in Chapel Hill, North Carolina, USA (coordinates 35.889114, −79.0145183), and sieved through a 2 mm sieve. Round 2-inch pots were filled with soil and sown with seeds of either *A. thaliana* Col-0 or *B. distachyon* BD21. Seeds were stratified for 2 days at 4 °C in the dark then placed into a growth chamber at 22 °C under short day conditions (8 h light 16 h dark) for 6 weeks.

**Harvest and DNA extraction.** After 6 weeks, plants were harvested, and shoots were separated from roots. Loosely attached soil was removed manually from the roots, and then the roots and the shoots were washed separately three times by vortexing 15 s in water. The washed tissues were dab dried, flash frozen and lyophilized. In addition, samples of 250 mg of soil were collected, flash frozen and lyophilized. DNA was extracted from the lyophilized samples using the Dneasy Powersoil Pro kit (QIAGEN).

**Sequencing library prep.** The V3V4 region of the 16S rRNA gene was amplified using a 2-step, dual-index approach to barcode the amplified DNA at the 3′ and 5′ ends[67]. Briefly, the V3V4 locus was amplified using Kapa Hifi (Kapa Biosystems) with primers 16S_V3 F and 16S_V4 R for 30 cycles. The products of this first PCR were then diluted 100-fold and used as template for the barcoding PCR with primers illumina_inx_x F and illumina_inx_x R for 10 cycles. Individual PCR libraries were pooled in equimolar ratios and cleaned using 0.8X AMPure XP beads (Beckman Coulter). Pooled libraries were sequenced on an Illumina MiSeq using 2 × 300 bp paired-end reads. We recovered on average 105,485 (range: 46,579–254,091) high quality bacterial sequences from each sample.

**Community assembly analysis.** We used the R package "DADA2" to process our sequencing reads[68], which infers amplicon sequence variants (ASV), and the R package "phyloseq" to further process our samples[69]. We constructed a phylogenetic tree of our ASVs with the phylogenetic placement method outlined in Janssen et al.[70] and implemented using the SILVA 128 reference alignment in QIIME2[71]. We simplified our dataset to include only common ASVs for downstream analyses of community composition. We defined common as occurring in at least 5 samples at an abundance of at least 25 sequences[13], which yielded 702 ASVs and accounted for 94% of the total number of sequences in the dataset. We performed a proportional abundance normalization (sequencing reads for an ASV in a given sample divided by the total number of sequencing reads in that sample) on this common set of ASVs for our analysis of community composition[69]. We also rarefied samples to the lowest read count in the dataset (46,579 reads) and found nearly identical results to the proportional abundance normalized analyses. We report results with the proportional abundance normalized dataset with the 5 × 25 threshold. We used permutational multivariate analysis of variance (R function "adonis2" from the "vegan" package v. 2.5-7, marginal significance of terms computed) to analyze bacterial composition using the common ASV dataset described above. We used three different measures of community distance, weighted UniFrac, unweighted UniFrac, and Bray–Curtis. We found highly significant effects of fraction for all samples, and host species for root and shoot samples regardless of the community distance measure used. For visualization purposes we performed principal coordinates analyses using unweighted UniFrac distance matrices.

### Gnotobiotic system, bacterial inoculum, and plants' growth
**Approach.** Bacterial mutant abundance on the plant and in the soil are quantified using amplicon sequencing of barcodes, Therefore, natural resident bacteria will not interfere with the identification of mutants. Yet, those resident bacteria can compete with members of the mutant library of a focal strain and reduce its colonization. Many colonizing bacteria are essential for successful analysis of differences in mutant fitness in the different environments. To enhance the colonization of members of the mutant library we followed techniques taken from gnotobiotic systems.

**Pot preparation.** Plants were grown in a Gnotopot system based on Kramer et al.[30]. In short, peat pellets were placed in 1 inch round pots and soaked in 0.5xMS until the pellets were saturated. The saturated pellets were then autoclaved three times, with 48 h incubation at room temperature between the first and second autoclave. The pots were distributed into flats and covered with tall domes.

**Seed preparation.** Seeds were sterilized for 5 min in 50% bleach, then were washed 3 times in sterile water. The sterile seeds were stratified for 3 days at 4 °C in the dark before sowing.

**Bacteria preparation.** Mutant libraries of *P. bryophila* MF376 and *P. simiae* WCS417 were grown overnight in 2xYT broth then washed and diluted in sterile water. A diverse auxiliary community of four plant-associated bacterial strains were also prepared in order to test the effect of competing/promoting strains. These strains, (Actinobacteria CurtobacteriumCL20, Alphaproteobacteria Sphingomonas MF220A, Betaproteobacteria Ralstonia CL21, and Gammaproteobacteria Stenotrophomonas MF92) were grown individually in 2xYT broth overnight, mixed to equal amounts, and the mixture was diluted in sterile water.

**Seed sowing and bacteria inoculation.** Each pot was sown with five stratified sterile seeds of either *A. thaliana* Col-0 or *B. distachyon* BD21. Half of the pots were inoculated with $10^8$ bacteria of the focal, mutant

library strain (either MF376 or WCS417), in isolation, while the other half was inoculated with $10^8$ bacteria from the focal strain accompanied with $10^7$ bacteria from the small auxiliary community. In each flat, two inoculated pots were left without plants as soil only controls.

**Plant growth.** The plants were transferred to a growth chamber in flats with plastic domes. They were grown for 6 weeks under short day conditions (8 h light, 16 h dark). After 2 weeks, the plants were thinned to three plants per pot. After 4 weeks, cracks in the domes were opened to reduce the humidity around the plants. The plants were grown for a total of 6 weeks before harvest.

**Harvest, preparation of sequencing libraries, and sequencing**
**Samples harvest.** Six-week-old plants were harvested. To increase the sample-to-sample reproducibility, the plants were pooled to groups of 4–5 pots per sample (12–15 plants total). The above (shoot) and below (root) the ground tissues were separated. To remove non-attached bacteria, the shoots and roots were washed three times as detailed above. The clean tissue was then ground using a mortar and pestle, then the ground tissue was emersed in 20 ml of antibiotic-supplemented 2xYT broth to enrich the bacteria within the plant, similar to previous work[25]. For soil samples, 1 g of soil was harvested for each sample and placed into antibiotic-supplemented 2xYT broth in a similar manner. The 2xYT medium was supplemented by antibiotic combinations that inhibit the growth of members of the auxiliary community but not of the focal strain: 100 mg/l Kanamycin and 4 mg/l Carbenicillin for MF376, and 100 mg/l Kanamycin and 8 mg/l Rifampicin for WCS417. After a short overnight growth (8 h), the DNA was extracted from the culture using DNeasy Power soil 96 kit (QIAGEN). Due to the high replication and the high number of samples, we had to split the harvest to 2 consecutive days.

**Sequencing library preparation.** Preparation of the libraries for sequencing was done following a 2-step, dual-index approach to barcode amplified DNA at the 3' and 5' ends: firstly a PCR to amplify the barcode region and add the internal parts of Illumina adapters, followed by a second PCR that added multiplexing indexes and the external parts of Illumina adapters. All amplification steps were done with KAPA HIFI readymix (KAPA biosystems). The first PCR with primers BS_int F and BS_int R (Supplementary Data 10) for 25 cycles amplified the barcodes and added adapters. The first PCR product was diluted 1:50 in water and a second PCR was performed with primers Illumina_inx_x F and illumina_inx_x R for 15 cycles to add multiplexing indexes and Illumina adapters. The libraries were then quantified on an agarose gel and pooled equimolarly. The pooled libraries were run again on 1.5% agarose gel for size selection, the fragments around 150 bp were cut from gel and cleaned with QIAquick Gel Extraction Kit (QIAGEN). The libraries were sequenced using 50 bp single reads on an Illumina Hiseq 2500.

**Sequencing analysis and identification of differentially abundant mutants**
**Basic analysis.** The assignment of fitness score to each mutant in each treatment was done based on the BarSeqR pipeline established by Wetmore et al.[21]. In short, the 20 bases barcodes were identified based on the preceding six nucleotide. The barcodes were then mapped to the bacterial genome using a mapping table. The BarSeqR pipeline calculates the fitness by comparing the mutants' abundance in each sample to its abundance in a T0 control. Since bacteria colonize the plant from the surrounding substrate, we defined the substrate samples (soil) as the control. The fitness of mutants in a gene is then computed as the $\log_2$ transform of a weighted average of all mutants in that gene[21]. The calculated fitness score of mutants in each gene for each sample is used for further analysis. Overall, we identified an average of 50,544 unique barcodes per sample, which sums up to ~8.5

unique barcodes per gene per sample on average (Supplementary Fig. 2).

**Identification of differentially abundant mutants.** The BarSeqR pipeline also computes a $P$ value based on $T$-like statistics from multiple competing mutants in the same gene. While that approach yields good results in culture with large inoculum[21], the limited number of plant colonization events leads to incomplete sampling from the diverse library and hence different starting population on each plant sample. Those differences in starting population hinder the use of that approach. Instead, we build a regression model for the fitness score of each mutant, comparing its fitness in the sample to its fitness in the soil: Fitness ~ Sample_type + Auxiliary_community + Harvest_day + (1| flat), where the Sample type (plant sample or soil), Auxiliary_community (Focal strain inoculated in isolation or with an auxiliary community) and Harvest_day (first or second day of harvest) are fixed variable and the flat number (overall we had 16 flats) is a random variable. Following the fit to the model, the distribution of residuals was tested, and genes that did not follow normal distribution in any of the samples were omitted from the analysis (Shapiro–Wilk test, $P$ value <0.05, a total of 486 genes). Genes that satisfy normal distribution and with a false discovery rate (FDR) corrected (Benjamini–Hochberg procedure) $P$ value <0.05 and $\log_2$(Fold change) > |1| were defined as "Affecting association with plants." The addition of the auxiliary community and the day of the harvest affected only a small number of genes and are not discussed in the paper.

**Engineer antibiotic-resistant derivatives of *P. bryophila* MF376**
**Approach.** Two derivatives of MF376 with differentiating antibiotic resistance were engineered to conduct a competition assay between WT and mutant derivatives. Antibiotic resistance cassettes were added to the genome in an intergenic region by a gene knock-in approach based on the SacB negative selection gene to create antibiotic-resistant parental derivatives Tet$^R$ WT and Chl$^R$ WT. All oligos sequences can be found in Supplementary Data 10.

**Cloning plasmids pMo130_par_ TetR and pMo130_par_ ChlR.** An intergenic region in the MF376 genome was identified at position 319,584–319,849 between genes H281DRAFT_00298 and H281DRAFT_00299. To clone antibiotics resistance genes into that region, we used the plasmid pMo130 that contains a Kanamycin resistance gene for positive selection and the SacB gene for negative selection. The plasmid was amplified by PCR using primers GD_pMo130 F and GD_pMo130 R. Fragments of 1000 bp upstream and downstream the intergenic point were picked from the MF376 genome by PCR using primers GD005_par_flkU F and GD006_par_flkU R (upstream) and GD007_par_flkD F and GD008_par_flkD R (downstream). The TetA gene that confers resistance to Tetracycline was picked from plasmid pBBR1-MCS3 with primers GD001_TetR R and GD002_TetR F. The Chloramphenicol acetyl transferase (CAT) gene that confers resistance to Chloramphenicol was picked from plasmid pBBR1-MCS1 with primers GD003_ChlR R and GD004_ChlR F. The primers that amplified the antibiotic resistance cassettes and the genomic fragments added complementary sequences on their 3' end. Each of the antibiotic resistance genes were cloned into plasmid pMo130 between the upstream and the downstream genomic fragments using Gibson assembly to create plasmids pMo130_par_TetR and pMo130_par_ChlR. Each of the new two plasmids were transformed to *E. coli* Top10 cells by heat shock transformation for propagation in 2xYT supplemented by 50 mg/l Kanamycin and extracted using QIAprep Spin Miniprep Kit (QIAGEN).

**Selecting antibiotic resistance derivatives of *P. bryophila* MF376.** Each of the plasmids pMo130_par_TetR and pMo130_par_ChlR were transformed into the 2,6-diaminopimelic acid (DAP) auxotrophic

biparental mating strain *E. coli* WM3064 by electroporation. The transformed mating strains were selected on LB supplemented by 0.3 mM DAP, 50 mg/l Erythromycin, and 50 mg/l Kanamycin. The transformed mating strains were each mixed with the parental strain MF376 and plated on LB agar plates supplemented with 0.3 mM DAP. After overnight growth, the transconjugated MF376 derivatives were selected on LB agar plates supplemented with 3 mg/l Tetracycline or 10 mg/l Chloramphenicol. The antibiotic-resistant mating strain derivatives were counter selected due to their DAP auxotrophy. Transconjugate MF376 clones were cleaned twice by streaking onto antibiotic LB agar plates, where the plasmid was able to integrate into the genome by homologous recombination and then plated on LB agar plates supplemented by 10% sucrose to select against the SacB gene. To survive the sucrose selection, bacteria must lose the integrated pMo130 plasmid and recombine either to the parental form (no antibiotic resistance cassette) or to the antibiotic resistance derivative (where the antibiotic resistance cassette resides between the genomic fragments). Next, colonies that were selected on 10% sucrose were streaked onto LB agar plates supplemented with 3 mg/l Tetracycline or 10 mg/l Chloramphenicol to select for the antibiotic-resistant derivatives Tet[R] WT and Chl[R] WT, respectively.

### Gene deletion in *P. bryophila* MF376

**Approach.** The ef90 operon was deleted from both antibiotic-resistant MF376 derivatives by a similar approach. All oligos sequences can be found in Supplementary Data 10.

**Cloning plasmid pMo130_d90.** Flanking DNA fragments (1 kb long) upstream and downstream of the ef90 operon were amplified from the genome of MF376 using primers GD055_flkU-3190 F and GD056_flkU-3190 R (upstream gene loci ID H281DRAFT_03188) and GD057_flkD-3190 F and GD058_flkD-3190 R (downstream gene loci ID H281DRAFT_03191). Primers GD056_flkU-3190 R and GD057_flkD-3190 F have 5' overhanging ends complementary to each other, and primers GD055_flkU-3190 F and GD058_flkD-3190 R have overhanging 5' ends complementary to the amplified pMo130 plasmid. The upstream and downstream genomic fragments were then integrated into the pMo130 plasmid by Gibson assembly to create plasmid pMo130_d90 that was then transformed into *E. coli* Top10 cells by heat shock transformation. The transformed Top10 cells were plated on LB agar supplemented by 50 mg/l Kanamycin, and after 2 days of growth colonies were validated using primers GD061_vo_3190 F and GD062_vo_3190 R that sit within the upstream and downstream MF376 genomic fragments, respectively. The plasmid was propagated in 2xYT supplemented by 50 mg/l Kanamycin and extracted using the QIAprep Spin Miniprep Kit (QIAGEN).

**Deleting operon *ef90* from MF376.** Plasmid pMo130_d90 was transformed into the biparental mating strain *E. coli* WM3064 by electroporation. The transformed derivative of the mating strain was selected onto LB agar plates supplemented by 0.3 mM DAP, 50 mg/l Erythromycin, and 50 mg/l Kanamycin. Next, the transformed derivative of the mating strain was mixed with each of the antibiotic-resistant parental derivatives Tet[R] WT and Chl[R] WT and plated onto LB agar plates supplemented with 0.3 mM DAP. After overnight growth, the transconjugated MF376 derivatives were selected on LB agar plates supplemented with 50 mg/l Kanamycin. Transconjugate Tet[R] WT and Chl[R] WT MF376 clones were cleaned twice by streaking on Kanamycin LB agar plates and then were plated onto LB agar plates supplemented with 10% sucrose to select against the SacB gene. Colonies growing on the sucrose plates were tested for the deletion of the ef90 operon by two PCRs. A reaction with primers GD061_vo_3190 F and GD062_vo_3190 R (within the upstream and downstream flanking fragments) yielded a band since the deleted operon left the flanking fragments in proximity to one another, and a reaction with primers

GD059_vi_3190 F and GD060_vi_3190 R (within the *ef90* operon) yielded no band indicating that the fragment was deleted. The new two derivatives, TetR d90 and ChlR d90, were cleaned by streaking on LB agar plates and revalidation by PCR.

### Competition assay between WT and d90 mutant derivative of *P. bryophila* MF376

**Macerated leaf extract preparation.** Plants were grown in a potting mix for 6 weeks under short day conditions (8 h light 16 h dark 22 °C/18 °C, respectively). After 6 weeks, plants were harvested the shoots and roots separated, and then weighed. Next, the shoots were mixed with equal volume of 2xYT broth and were ground using a mortar and pestle. The macerated leaf extract was filtered through a 0.22 μm filter to remove bacteria, serial diluted in 2XYT broth and 90 μl was pipeted into 96-well flat bottom plates.

**Bacterial preparation.** Each of the four MF376 derivatives was cultured shaking overnight at 28 °C in 2xYT. All derivatives were diluted to $OD_{600}$ 0.01 and WT derivatives were mixed with mutant derivatives as follows: ChlR WT was mixed with mutant TetR d90 and reciprocally TetR WT was mixed with mutant ChlR d90.

**Competition assay, plating, and imaging.** The mixed bacterial cultures (ChlR WT versus TetR d90 and TetR WT versus ChlR d90) were diluted 1:10 into the macerated soot extract (10 μl into 90 μl) to yield an inoculum of $10^5$ bacteria. After overnight growth in a shaking incubator at 28 °C, the culture was serial diluted in water and 5 μl drops from each dilution of each culture were plated onto LB agar plates supplemented with either 8 mg/l Chloramphenicol or 3 mg/l Tetracycline. Colonies were grown for 2 days at 28 °C and the plates were imaged using a tabletop scanner. The binary nature of the growth (full growth inhibition vs full growth) negated the need for quantification.

### Comparing the growth of WT *P. simiae* WCS17 and an *ef90* homolog mutant in macerated leaf extract

Both WT WCS417 and the *ef90* mutant were cultured shaking overnight at 28 °C in 2xYT. The bacterial cultures were diluted to $OD_{600}$ 0.01 and further diluted 1:10 into serial dilutions of macerated leaf extract (10 μl into 90 μl) to yield an inoculum of $10^5$ bacteria in each well of a flat bottom 96-well plate. Bacteria were grown overnight shaking at 28 °C in a temperature-controlled plate reader (Infinite 200 pro, Tecan). During growth, $OD_{600}$ was measured every 10 min. Plotting bacterial growth over time, we defined the time when the no-plant control culture reached 90% culture saturation as the endpoint of the experiment. For each sample, the final OD600 was defined as the OD600 at the endpoint. Background OD600 was defined as the fifth lowest OD600 measurement (absorbance of plant material diminishes over time so taking initial OD600 yields negative values). The reported OD600 is final OD600 minus background OD600.

### Quantifying ef90 expression in A.t. leaves and in macerated leaf extract

**Infiltration and harvest.** Overnight culture of WT MF376 and WCS417 were diluted 1:20 in 2xYT and recovered for 2 h shaking at 28 °C. The recovered bacteria were diluted to OD600 0.01, washed twice in 10 mM $MgCl_2$, and hand infiltrated into leaves of 6-week-old *A. thaliana* Col-0. Leaves were harvested after 15 and 60 min. As a control, leaves infiltrated with $MgCl_2$ without bacteria were sampled. All samples were flash froze upon harvest.

**Inoculation in macerated leaf extract and harvest.** Macerated leaf extract was diluted 1:4 in 2xYT. Log phase bacteria were inoculated into the diluted leaf extract at a final $OD_{600}$ of 0.01. After 15 min, the cells were pelleted by centrifugation at $5000 \times g$ for 2 min, and the supernatant was removed in the cells were flash frozen in liquid nitrogen.

**RNA extraction, RT-qPCR, and analysis.** RNA was isolated using RNeasy kit (QIAGENE), and cDNA was prepared using Superscript IV (Invitrogen). The qPCR was preformed using PowerUp SYBR Green (Applied Biosystems) on a VIIA7 machine (Applied Biosystems). Relative gene expression was calculated as $-\Delta\Delta CT$ with *recA* as the internal control. There was no signal in "no bacteria" control samples. All primers used for RT-qPCR can be found in Supplementary Data 10.

**Building a gene tree of all efflux pumps in a 175-member isolate-collection**
The inner-membrane subunit of all Gram-negative members of the 185-member isolate collection was identified by the COG identifier COG0841 (Multidrug efflux pump subunit AcrB). Overall, 823 genes were identified. All pairwise protein sequences were aligned and the pairwise distances were measured as a BLOSUM50 dissimilarity score using the seqpdist command (MATLAB R2024a). Next, a neighbor-joining gene tree of all 823 genes were built based on the distance matrix. The tree was plotted using ggtree (R). Shoot clade (ef90 orthologs) and root clade were identified based on the most recent common ancestor of the functionally tested genes from MF376 and WCS417.

**Measure the toxicity of selected isothiocyanates**
The commercially available isothiocyanates Sulforaphane (4MSO-ITC, Sigma, Cat. S6317) and Iberin (3MSO-ITC, Sigma, Cat. SML3844) were dissolved in dimethyl sulfoxide to a stock concentration of 20 mg/ml. Each of the two compounds were serially diluted 1.5-fold in 2xYT in a microtiter plate. The diluted compounds were inoculated by $5 \times 10^5$ bacteria per well, and plates were incubated overnight shaking at 28 °C. After overnight growth, bacterial load was estimated by OD600.

**Reporting summary**
Further information on research design is available in the Nature Portfolio Reporting Summary linked to this article.

## Data availability
All sequencing data generated and analyzed in the current study are deposited in publicly available archives. The 16S data is deposited at the sequencing read archive (SRA) as BioProject PRJNA1189262 and the BarSeq barcodes sequencing is deposited at the European nucleotide archive (ENA) as study ERP166428 [https://www.ebi.ac.uk/ena/browser/view/PRJEB82761]. The TnBarSeq results are available at Figshare (https://doi.org/10.6084/m9.figshare.28716422). All other data and scripts are available upon request.

## Code availability
Figures were generated with custom R and MATLAB codes. All custom codes used in this study are available upon request.

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

## Acknowledgements

This work was supported by NSF grant IOS-1917270 to J.L.D. and C.D.J. J.L.D. is an Investigator of the Howard Hughes Medical Institute, supported by the HHMI. D.R. gratefully acknowledges funding by EMBO Long Term Fellowship (ALTF 743–2019). C.R.F. gratefully acknowledges funding by a Natural Sciences and Engineering Research Council of Canada postdoctoral fellowship (532852-2019). We thank Rex Malstrom and Adam Deutschbauer, DOE-JGI and DOE-LBNL, for providing bacterial strains used in this work. We thank Dr. Detlef Weigel, Dr. Jonathan M. Conway, Dr. Trent R. Northen, and Dr. Katherine Louie for technical discussions. We thank Dr. Omri Finkel, Dr. Karnelia Paul, Dr. Adam Rosenthal, and Dr. Sarah R. Grant for critical comments and reading of the manuscript and the Dangl lab microbiome team for comments throughout the course of this project. This article is subject to HHMI's Open Access to Publications policy. HHMI lab heads have previously granted a nonexclusive CC BY 4.0 license to the public and a sublicensable license to HHMI in their research articles. Pursuant to those licenses, the author-accepted manuscript of this article can be made freely available under a CC BY4.0 license immediately upon publication.

## Author contributions

D.R., C.R.F., and J.L.D. conceived and designed the experiments. D.R., C.R.F., C.S., and T.L. performed the experiments. D.R., C.R.F., D.J.K., and C.D.J. analyzed the data. D.R. and J.L.D. wrote the manuscript with feedback from all authors.

## Competing interests

The authors declare no competing interests.
