## [Transparent Peer Review file · Nature Communications]

An effluent pump family distributed across plant commensal bacteria conditions host- and organ-specific glucosinolate detoxification

Corresponding Author: Dr Jeffery Dangl

Version 0:

Reviewer comments:

Reviewer #1

(Remarks to the Author)

I have previously reviewed this manuscript for another journal. In this new version, authors have carefully addressed most of the major issues raised during the first evaluation round.

Although the concerns regarding in planta validation and the ecological relevance of the phenomenon have been only partially addressed, the work appears solid, novel and the conclusions are supported by genetic evidence. Therefore, I would recommend this work for publication in this journal.

Reviewer #2

(Remarks to the Author)

Russ et al. have addressed many concerns raised and added new data (including data on glucosinolate quantification in Arabidopsis mutants, all the data generated and more details for certain experiments) and modifications in the text. This will ensure transparency and reproducibility of results.

Nevertheless, one point has not been properly addressed in my opinion, even though two reviewers raised it: the extension of the results of this study to other species, including those of agronomic interest (Broccoli, for instance). I'm not entirely convinced by the argument that "this is like asking a paper on mouse genetics to repeat the observation in another distantly related mammal". As mentioned in the conclusion, one of the long-term aims of these results is to address "the challenge for application of microbial interventions across different cultivars and across different crop species" so beyond the Arabidopsis genus. I agree that we can't be sure of getting the same trend on other plants due to a specific plant-bacteria interaction/adaptation, so I don't think it's necessary to perform these new experiments within the scope of this article, which is already gigantic in terms of the number of experiments and results, but the authors should mention perhaps line 565 that this result can be extended in the future to plants producing the same glucosinolate variants.

I am convinced that the results of this study will be an excellent addition to the field.

Reviewer #3

(Remarks to the Author)

I appreciate the efforts made by the authors to address the many comments raised by the three reviewers. I do not have further suggestions.

We would like to thank all the reviewers for their support of the publication of our manuscript **“An effluent pump family distributed across plant commensal bacteria conditions host- and organ-specific glucosinolate detoxification”**. We would especially appreciate the detailed comments that helped improve our manuscript.

REVIEWERS' COMMENTS

Reviewer #1 (Remarks to the Author):

I have previously reviewed this manuscript for another journal. In this new version, authors have carefully addressed most of the major issues raised during the first evaluation round.

Although the concerns regarding in planta validation and the ecological relevance of the phenomenon have been only partially addressed, the work appears solid, novel and the conclusions are supported by genetic evidence. Therefore, I would recommend this work for publication in this journal.

Thank you for the appreciation and support for the MS.

Reviewer #2 (Remarks to the Author):

Russ et al. have addressed many concerns raised and added new data (including data on glucosinolate quantification in Arabidopsis mutants, all the data generated and more details for certain experiments) and modifications in the text. This will ensure transparency and reproducibility of results.

Nevertheless, one point has not been properly addressed in my opinion, even though two reviewers raised it: the extension of the results of this study to other species, including those of agronomic interest (Broccoli, for instance). I'm not entirely convinced by the argument that “this is like asking a paper on mouse genetics to repeat the observation in another distantly related mammal”. As mentioned in the conclusion, one of the long-term aims of these results is to address “the challenge for application of microbial interventions across different cultivars and across different crop species” so beyond the Arabidopsis genus. I agree that we can't be sure of getting the same trend on other plants due to a specific plant-bacteria interaction/adaptation, so I don't think it's necessary to perform these new experiments within the scope of this article, which is already gigantic in terms of the number of experiments and results, but the authors should mention perhaps line 565 that this result can be extended in the future to plants producing the same glycosinolate variants.

I am convinced that the results of this study will be an excellent addition to the field.

Thank you for the appreciation and support for the MS. We, too, believe that our findings can be tested in the future on other crops, and as you suggest we added a note on that in the discussion section (now row 422).

Reviewer #3 (Remarks to the Author):

I appreciate the efforts made by the authors to address the many comments raised by the three reviewers. I do not have further suggestions.

Thank you for your appreciation and support for the MS.